# Variation in Pathological Appearance Across Repeated Sampling from Probably Benign Breast Lesions

**DOI:** 10.3390/biomedicines13122897

**Published:** 2025-11-27

**Authors:** Athanasios Zouzos, Irma Fredriksson, Theodoros Foukakis, Johan Hartman, Fredrik Strand

**Affiliations:** 1Department of Radiology, Karolinska University Hospital, 17176 Stockholm, Sweden; irma.fredriksson@ki.se (I.F.); theodoros.foukakis@ki.se (T.F.); fredrik.strand@ki.se (F.S.); 2Department of Oncology and Pathology, Karolinska Institute, 17164 Stockholm, Sweden; johan.hartman@ki.se; 3Department of Molecular Medicine and Surgery, Karolinska Institute, 17164 Stockholm, Sweden; 4Department of Pathology, Karolinska University Hospital, 17176 Stockholm, Sweden

**Keywords:** breast, vacuum-assisted excision, core-needle biopsy, fine-needle aspiration, B3 lesions

## Abstract

**Background:** The diagnostic process for probable benign breast lesions involves a 1–40% upgrade rate to malignancy when biopsy (cytology and/or histology) is compared with surgery. In a previously conducted clinical randomized trial, we aimed to examine diagnostic discrepancies between prior biopsy results and subsequent vacuum-assisted excision (VAE). **Methods**: This study is a post hoc analysis of the Swedish VAE randomized trial. Patients were enrolled between November 2019 and August 2022. All patients who underwent a biopsy before VAE were included in this study. Pathology reports from the initial biopsy, VAE, surgical excision, and recurrence were collected. In addition, we conducted clinical follow-up, including imaging, for at least 2 years. **Results**: The study population included 169 patients with 169 lesions, of whom 71 underwent fine-needle aspiration cytology (FNA), and 126 underwent core-needle biopsy (CNB) before VAE. The diagnostic discrepancy between FNA and VAE was 38% (27/71). The discrepancy between CNB and VAE was 29% (37/126). The upgrade rate to cancer was 7% (5/71) for FNA and 5% (6/126) for CNB. In the CNB group, the highest upgrade rate to cancer occurred in patients with prior atypical ductal hyperplasia (ADH) on CNB (3/12, 25%). **Conclusions**: The upgrade rate in histopathological diagnosis between prior CNB and VAE was high (15%), and even higher when comparing FNA with VAE (24%). Our findings support avoiding FNA for BI-RADS 3 and 4a lesions and suggest that multi-round VAE may be a safe and effective alternative to surgery for selected cases, particularly those with ADH on CNB.

## 1. Introduction

Several proliferative, non-proliferative, and non-malignant breast changes are associated with the risk of developing cancer [1,2]. To diagnose these changes and determine their prognostic significance, samples from core-needle biopsy (CNB) and/or fine-needle aspiration (FNA) [3], with needle sizes varying from 27G (FNA, outer diameter 0.4 mm) to 12G (CNB), were analyzed in comparison to surgical samples.

FNA is less and less used in modern breast diagnostics, limited by its inability to distinguish between in situ and low-grade cancers or lobular carcinoma. Furthermore, FNA cannot differentiate between sclerosing and papillary lesions or atypical ductal hyperplasia (ADH) [4,5]. It can, however, be used without local anesthetics, and a report can usually be generated within a few hours. The categorization of cytological features is generally performed using the classification proposed by Wells et al. (C1–C5) [6].

CNB is considered the preferred choice over repeated FNA [7] and the categorization of histological features is performed using the B coding (B1–B5) [8]. However, even CNB has its limitations, including an upgrade rate to invasive cancer of up to 20% following an initial in situ diagnosis, and an upgrade rate of B3 lesions to malignancy ranging from 3 to 40%, depending on the lesion type [9].

Diagnostic techniques such as FNA, CNB are determined according to radiological features, which are generally categorized using the Breast Imaging Reporting and Data System (BI-RADS) [10]. Although the characterization of breast lesions according to BI-RADS can be challenging [11], it is generally accepted that most of the indeterminate lesions (C3 and B3) have imaging characteristics that correspond to BI-RADS 3 and BIRADS 4a categories.

Over the past two decades, larger-diameter biopsy needles, up to 7G (outer diameter 4.6 mm), have been introduced and paired for use in vacuum-assisted biopsy (VAB), resulting in reduced rates of histological underestimation [9,12,13]. The primary advantage of VAB, compared to CNB, besides its larger needle size, is its ability to obtain multiple samples through a single small skin incision, thereby enhancing tissue sampling and diagnostic accuracy [14,15]. For certain lesions, continued sampling until the entire lesion is no longer visible has been established as an alternative to open surgical excision (OSE), a technique known as vacuum-assisted excision (VAE).

Histopathological findings are categorized according to the European guidelines for quality assurance of breast cancer screening and diagnosis [8,16]. These categories include B1 (normal or non-diagnostic tissues), B2 (benign lesions), and B3 (lesions with a 9.9–35.1% overall risk of upgrading to malignancy upon subsequent surgery) [17], B4 (suspected malignant lesions), and B5 (malignant lesions). The B3 category, referred to as “probably benign,” is the most controversial in therapeutic management and includes: ADH, flat epithelial atypia (FEA), classical lobular neoplasia (LN) (including atypical lobular hyperplasia (ALH) and classical lobular cancer in situ (cLCIS), phyllodes tumors, intraductal papillomas (with or without atypia), radial scars or complex sclerosing lesions, mucocele-like lesions, and other miscellaneous lesions [16,18]. The prevalence of B3 lesions in otherwise healthy women ranges from 5 to 10% [19,20,21]. Most B3 lesions represent a clinical dilemma, as the only definitive method of excluding malignancy is OSE, often regarded as the “gold standard” for conclusive diagnosis [17]. Given the advantages of improved patient comfort and cost-effectiveness, utilizing VAE for managing B3 lesions has become increasingly common [22]. Since 2016, several organizations [23,24,25,26] have endorsed VAE for the management of B3 lesions. 

This study aims to identify the patterns of pathological diagnoses across sequential diagnostic procedures in individual patients, including initial biopsy, successive VAE sampling containers, and any follow-up diagnoses within 2 years, to enhance our understanding of B3 lesions and inform strategies for optimized diagnosis and personalized management. 

## 2. Materials and Methods

### 2.1. Trial Population

This retrospective study was based on the Swedish VAE trial, whose primary endpoints were procedure time and completeness of breast lesion removal [27]. The present study focused on histopathological reports across the diagnostic pathways of each patient, from CNB and/or FNA, and across three sequential tissue sample containers collected during VAE. From the established study population of the Swedish VAE trial, we included patients with ultrasound-visible lesions < 30 mm, corresponding to BI-RADS 3 and 4A [10], and excluded those who had not undergone a biopsy prior to VAE.

### 2.2. Diagnosis and Excision Procedure

Lesions were diagnosed before VAE using ultrasound-guided CNB (14G needle) and/or FNA (22G needle). For lesion excision, the EnCor EnSpire™ Breast Biopsy System (Becton, Dickinson and Company (BD), Franklin Lakes, NJ, USA), primarily used for biopsies in our department, was employed. Microcalcifications were excised using a stereotactic technique with mammographic guidance, while all other lesions were excised using ultrasound guidance. The goal was to excise each lesion in three sequential rounds of VAE, where the tissue samples were placed in three numbered containers. The first container included all tissues from the area where no remaining lesions were visible on mammography or ultrasound. The second container included a 360-degree biopsy sweep within the cavity of the first container, while the third container included an additional 360-degree biopsy sweep, further expanding the excision cavity.

### 2.3. Data Analysis

The primary outcome was diagnostic discrepancy between the initial biopsy (CNB or FNA) pathology report and the results from successive tissue containers at VAE, and any follow-up diagnosis within 2 years (same breast location).

Histopathological analysis followed the European guidelines for quality assurance of breast cancer screening and diagnosis [8,16], where lesions were categorized as follows: B1: normal breast tissue; B2: fibroadenomas/fibroadenosis, hemangiomas, and pseudoangiomatous stromal hyperplasia (PASH); B3: papillomas, radial scars or complex lesions, FEA, ADH, and LN; and B5: ductal cancer in situ (DCIS) or invasive cancer. For cytological analysis, due to the inherent diagnostic limitations following FNA [4], we utilized the four categories recommended by Wells et al. [6]; C1: normal breast tissue, C2: fibroadenosis, C3: papillary formations, and C4: atypia, where malignancy could not be excluded.

### 2.4. Statistical Analysis 

The prevalence of each lesion was summarized and stratified by diagnostic and treatment methods. Descriptive statistics for each lesion type were calculated for relevant continuous and categorical parameters. All analyses were conducted using Stata/IC version 16.1 (StataCorp LLC, College Station, TX, USA), with statistical significance set at *p* < 0.05. 

## 3. Results

### 3.1. Study Population 

A total of 208 patients were screened and enrolled between November 2019 and August 2022 after obtaining written informed consent. Following eligibility assessment and withdrawals, the final trial population comprised 194 patients with one lesion per patient, of whom 169 had undergone CNB and/or FNA before VAE. The mean patient age was 50.1 years, and the mean lesion size was 10.4 mm. Table 1 summarizes the characteristics of the study population. 

### 3.2. Change in Primary Diagnosis After FNA

Seventy-one patients had an initial FNA report. When we compared the FNA diagnosis with the final diagnosis after VAE, we found that 44 primary diagnoses (61%) remained unchanged, 17 (24%) were upgraded, and 10 (14%) were downgraded (Table 2).

Of the 10 lesions initially diagnosed as normal tissue on cytology, 8 (80%) were upgraded—4 to fibroadenosis, 3 to papilloma without atypia, and 1 to myofibroblastoma.

Of the 11 lesions diagnosed as fibroadenosis on FNA, 2 (18%) were upgraded: 1 to papilloma without atypia and 1 to radial scar/sclerosing adenosis.

Of the 40 lesions with papillary formations on FNA, 30 (74%) were later diagnosed as papilloma without atypia on CNB or VAE. Five (10%) were upgraded to ADH (2 lesions), LN (1 lesion), and DCIS grade I (2 lesions).

Regarding atypia diagnoses, two lesions were upgraded to invasive cancer, while the rest remained unchanged or were downgraded. 

### 3.3. Change in Diagnosis After CNB

A total of 126 patients were diagnosed based on CNB, including 28 who underwent initial FNA. After histopathological analysis of VAE tissue, 89 lesions (71%) retained their initial diagnoses, 19 (15%) were upgraded, and 18 (14%) were downgraded (Table 3).

In the B1 category, 62% (5/8) were upgraded; three cases were upgraded to papilloma without atypia, and two to radial scar/sclerosing adenosis. In the B2 category, 13% (5/37) were upgraded to B3, including one case to ALH, two to radial scar/sclerosing adenosis, and two to papilloma. In the B3 category, 10% (8/79) were upgraded: five to cancer (two to invasive cancer and three to DCIS grade I), and three to increased associated atypia (B4). In the B5 category, of the two lesions diagnosed as DCIS grade I, one was upgraded to DCIS grade II, and one was downgraded to ADH.

### 3.4. Comparison of Diagnoses Between Subsequent VAE Rounds

Of the 169 patients who underwent one round of VAE; 115 underwent a second round (68%); and 42 underwent a third round (26%). Due to technical issues, 11 lesions from the second-round group and four from the third-round group could not be discriminated from the first round. Consequently, the second-round group included 104 diagnoses, and the third-round group included 38.

Table 4 shows the comparison between the diagnoses after the subsequent VAE rounds. Of the 104 lesions in the second round, 96 (92%) retained their initial diagnosis or were classified as normal tissue. The largest discrepancy was observed in the B1 category, where two lesions (50%) were upgraded to papillomas in the third round. Additionally, five lesions (31%) in the B1 category were upgraded after the second round: two to papilloma, two to ADH, and one to LN. In total, 8/104 (8%) lesions were upgraded in the second round, and 2/38 (5%) upgraded in the third round.

### 3.5. Comparison of Diagnoses Between VAE and Surgery

A total of 13 patients underwent open surgery as the final treatment following decision-making during a multidisciplinary conference (Table 5). Of these, seven patients were diagnosed with cancer after VAE (DCIS or invasive carcinoma), one with ADH, two with papilloma without atypia, one with fibroadenoma, one with radial scar/complex lesion, and one with adenomyolipoma. 

One of the six (17%) non-malignant lesions was upgraded to malignancy after surgery from ADH to DCIS grade II. Three patients had only scar tissue in the specimen after the previous VAE, while the remaining 9/13 (69%) had residual tissue from the targeted lesion, with no change in their diagnosis.

### 3.6. Cancer Diagnoses

A total of 10 cancer diagnoses were made after VAE and/or subsequent surgery. 

The highest upgrade rate occurred in patients with ADH on CNB (3/12, 25%), followed by ADH diagnosed during VAE (1/5, 20%), ALH on CNB (1/5, 20%), atypia on cytology (2/10, 20%), papillary formations on cytology (3/40, 8%), and papilloma without atypia on CNB (2/50, 4%). Two patients had suspicious DCIS grade I after CNB, but only one maintained this diagnosis after VAE. 

Changes in diagnoses among the different categories during subsequent biopsies are presented in a Sankey diagram (Figure 1).

## 4. Discussion

In this post hoc analysis of the Swedish randomized VAE trial, we found substantial diagnostic discrepancies between initial breast biopsy and subsequent VAE in lesions with uncertain malignant potential. The discrepancy rate was 29% for CNB and 38% for FNA. ADH on CNB was associated with the highest malignancy upgrade rate (25%). These findings raise important considerations for biopsy selection and follow-up strategies in the clinical management of B3 lesions.

Previous studies, including one by Deb et al. [28], have explored diagnostic discrepancies in benign and indeterminate breast lesions, though few have directly compared FNA and CNB to VAE. In our cohort, 8 of 10 lesions initially categorized as C1 on FNA were upgraded after VAE, reflecting key limitations of FNA such as inadequate sampling—often due to lesion consistency [5]—and interpretive variability [4]. These limitations can result in underdiagnosis, unnecessary repeat procedures, and potentially delayed treatment. Our findings reinforce the need to move away from FNA in favor of more reliable biopsy methods.

For CB, the overall upgrade rate was 15%, with the highest discrepancy observed in B1 lesions (62%), likely due to similar sampling limitations as FNA [29]. Prior studies have suggested that diagnostic accuracy in CNB improves with increasing numbers of cores [30], but we were unable to assess this due to a lack of documentation on core counts or needle gauge. Downgrade rates were 14% for both FNA and CNB, in line with previous reports [28], indicating that both under- and over-classification remain common with standard biopsy.

The trial protocol included a three-round VAE procedure, which previously showed higher complete excision rates [27]. In this study, we also observed improved diagnostic concordance with multiple rounds. Notably, our only case of malignancy following VAE occurred in a patient with ADH on CNB who underwent only a single VAE round. This underscores the value of multiple VAE passes not only for therapeutic purposes but also for diagnostic reliability. Our results support the growing view that surgery may be safely omitted in selected ADH cases, provided that VAE is performed thoroughly, with at least three excision rounds.

ADH remains the B3 lesion with the highest and most variable reported upgrade rate (5–50%) [23]. Our observed rate of 22% is consistent with prior literature and highlights the heterogeneity across studies. Variability in inclusion criteria, interobserver diagnostic reproducibility [31] and biopsy technique (e.g., core number, needle size) may all contribute to this range. Our findings support targeted excision of ADH using multi-round VAE as a minimally invasive alternative to surgery, aligning with a risk-adapted approach to care.

Papillomas diagnosed on CNB were the most common B3 lesion in our cohort. The observed upgrade rate to malignancy was low (4%), with only one case each of DCIS grade I and invasive carcinoma. Only one papilloma had associated atypia (ADH). These results are consistent with earlier studies reporting malignancy upgrade rates of 1.7–12% for papillomas without atypia [32,33] and support current guidelines favoring imaging surveillance after VAB in cases of radiologic–pathologic concordance.

Our study has several limitations. First, there is potential for indication bias, as the initial biopsy method (FNA vs. CNB) was selected by the diagnosing radiologist [34], possibly influenced by lesion characteristics not captured in this study. This could underestimate the true discrepancy of FNA if simpler lesions were preferentially selected for that method. Second, subgroup analyses were limited by the low frequency of some lesion types, such as lobular neoplasia, radial scars, and ADH. Third, we did not capture lesion morphology or imaging features, which are known to influence upgrade risk. Finally, this was a single-center study, although our results are consistent with international literature.

## 5. Conclusions

In an era that prioritizes minimally invasive diagnostics and personalized care, our findings offer strong evidence to guide clinical decision-making. FNA is insufficiently reliable and should be avoided for BI-RADS 3 and 4a lesions. CNB, while more accurate, still carries a non-negligible risk of diagnostic discrepancy, especially in ADH. For ADH, multi-round VAE appears to provide both diagnostic and potential therapeutic value, offering a promising alternative to surgery. Papillomas without atypia may be safely managed with imaging follow-up if radiologic-pathologic concordance is confirmed. Future studies should focus on refining criteria for VAE-based management, incorporating imaging and lesion morphology to support individualized, evidence-based care.

## Figures and Tables

**Figure 1 biomedicines-13-02897-f001:**
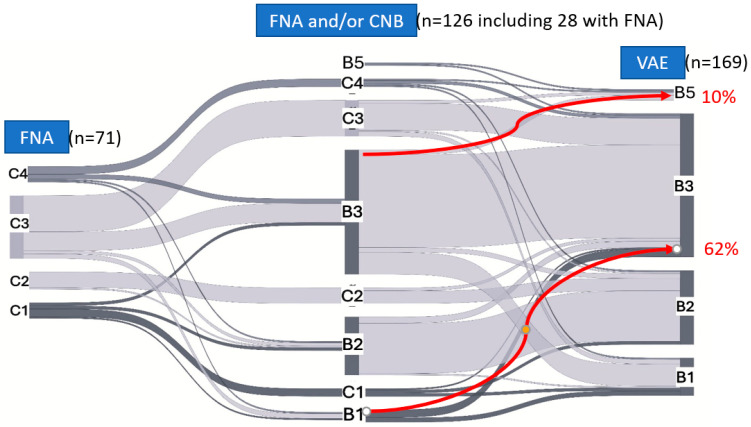
Sankey graph showing the changes in diagnoses during subsequent biopsies. FNA diagnoses to the left; CNB and FNA without subsequent CNB in the middle; VAE to the right; C1–C5 represent the cytological classification while B1–B5 the histological one. The red arrows show the most significant upgrade flows (from B1 to B3 lesions (62%) and from B3 to B5 (10%)). FNA, fine-needle aspiration; CNB, core-needle biopsy; VAE, vacuum-assisted excision.

**Table 1 biomedicines-13-02897-t001:** Study population characteristics.

	Total Number of Lesions (*n* = 169)
FNA only	43 (25%)
CNB only	98 (58%)
FNA and CNB	28 (17%)
VAE 1 round only	54 (32%)
VAE 2 rounds only	73 (43%)
VAE 3 rounds	42 (25%)
Surgery	13 (7.7%)
24-month follow-up	133 (79%)

FNA, fine-needle aspiration; CNB, core-needle biopsy; VAE, vacuum-assisted excision.

**Table 2 biomedicines-13-02897-t002:** Comparison between FNA and VAE (*n* = 71).

	*n*, (%)	Upgraded	Downgraded	Unchanged
Normal tissue	10 (14%)	8 (80%)	−	2 (20%)
Fibroadenosis	11 (15%)	2 (18%)	−	9 (82%)
Papillary formations	40 (56%)	5 (13%)	5 (13%)	30 (74%)
Atypia	10 (14%)	2 (20%)	5 (50%)	3 (30%)

FNA, fine-needle aspiration; VAE, vacuum-assisted excision.

**Table 3 biomedicines-13-02897-t003:** Comparison between CNB and VAE (*n* = 126).

	*n* (%)	Upgraded	Downgraded	Unchanged
B1	8 (6%)	5 (62%)	−	3 (38%)
B2	37 (29%)	5 (13%)	1 (3%)	31 (84%)
B3				
Papillomas	51(40%)	4 (8%)	4 (8%)	43 (84%)
Radial scar/complex lesion	6 (5%)	1 (17%)	3 (50%)	2 (33%)
ADH	12 (9%)	2 (17%)	6 (50%)	4 (33%)
Low-grade LN	5 (4%)	1 (20%)	−	4 (80%)
FEA	4 (3%)	−	3 (75%)	1 (25%)
Myofibroblastoma	1 (1%)	−	−	1 (100%)
B5 (DCIS I)	2 (2%)	1 (50%)	1 (50%)	−

CNB, core-needle biopsy; VAE, vacuum-assisted excision; ADH, atypical ductal hyperplasia; LN, lobular neoplasia; FEA, flat epithelial atypia; DCIS I, ductal cancer in situ grade 1.

**Table 4 biomedicines-13-02897-t004:** Comparison between the different first-round VAEs with the second (*n* = 104) and third (*n* = 38) rounds.

Comparison Between 1st and 2nd Round			Comparison Between 1st and 3nd Round		
Round	Diagnosis	*n* (%)	Round	Diagnosis	*n* (%)
1st	B1	16	1st	B1	4
2nd	Unchanged/Normal tissue	10 (63%)	3rd	Unchanged/Normal tissue	2 (50%)
	Upgraded	6 (37%)		Upgraded	2 (50%)
1st	B2	32	1st	B2	11
2nd	Unchanged/Normal tissue	31 (97%)	3rd	Unchanged/Normal tissue	11 (100%)
	Upgraded	1 (3%)		Upgraded	0
1st	B3	53	1st	B3	21
2nd	Unchanged/Normal tissue	52 (98%)	3rd	Unchanged/Normal tissue	21 (100%)
	Upgraded	1 (2%)		Upgraded	0
1st	B5	3	1st	B5	2
2nd	Unchanged/Normal tissue	3 (100%)	3rd	Unchanged/Normal tissue	2 (100%)

**Table 5 biomedicines-13-02897-t005:** Comparison between final VAE diagnosis and postoperative diagnosis (*n* = 13).

VAE Diagnosis	Postop Diagnosis	Result
Radial scar/complex lesion	Radial scar/complex lesion	Unchanged
Invasive carcinoma	Invasive carcinoma	Unchanged
Invasive carcinoma	Invasive carcinoma	Unchanged
Invasive carcinoma	Scar tissue	Downgraded/normal
Fibroadenoma	Fibroadenoma	Unchanged
DCIS II	Invasive carcinoma	Unchanged
DCIS I	DCIS III	Unchanged
DCIS I	DCIS I	Unchanged
DCIS I	Scar tissue	Downgraded/normal
Papilloma without atypia	LN	Unchanged
Papilloma without atypia	Papilloma without atypia	Unchanged
ADH	DCIS II	Upgraded
Adenomyolipoma	Scar tissue	Downgraded/normal

VAE, vacuum-assisted excision; ADH, atypical ductal hyperplasia; LN, lobular neoplasia; DCIS I–III, ductal cancer in situ grade 1–3.

## Data Availability

The data presented in this study are available on request from the corresponding author due to legal and ethical reasons.

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
