# Peer review of "Variation in Pathological Appearance Across Repeated Sampling from Probably Benign Breast Lesions"

_biomedicines, 2025, doi:10.3390/biomedicines13122897_

Round 1

Reviewer 1 Report

Comments and Suggestions for Authors

Core biopsy is mandatory for diagnosis of breast cancer and to detect molecular subtypes and for management so your research can focus on this

We should do our best to improve core biopsy as it is mandatory to Neo adjuvant chemotherapy 

Immunohistochemistry should be used to improve diagnosis of core biopsy 

Author Response

Comment: Core biopsy is mandatory for diagnosis of breast cancer and to detect molecular subtypes and for management so your research can focus on this

Resonse: We completely agree. Although fine-needle aspiration is still practiced in certain departments, the intention of articles such as ours is to emphasize that core biopsy is the better diagnostic method.

Reviewer 2 Report

Comments and Suggestions for Authors

1) What is the main question addressed by the research?

- The presented paper is devoted to the comparative analysis of diagnostic significance of two biopsy methods: fine- 20 needle aspiration cytology (FNA) and core-needle biopsy (CNB). The authors concluded that CNB demonstrated the highest update rate to cancer.

2) Do you consider the topic original or relevant to the field? Does it address a specific gap in the field? Please also explain why this is/ is not the case.

- The paper seems to be relevant to the scope of the journal. However, the results present only data analysis of moderate scale without any experimental work. Therefore, this manuscript could better fit to short communication format.

3) What does it add to the subject area compared with other published material?

- The subject area lacks novelty as comparison between fine- 20 needle aspiration cytology (FNA) and core-needle biopsy (CNB) were already published [PMID: 25174291]

4) What specific improvements should the authors consider regarding the 
methodology? 

- Authors should reformulate the conclusions to reach the necessary level of scientific novelty.

5) Are the conclusions consistent with the evidence and arguments presented and do they address the main question posed? Please also explain why this is/is not the case.

- The manuscript lacks practical conclusions in, for example, routing of the patients with breast lesions in clinic.

7)  Are the references appropriate?

- The references are appropriate

8)  Any additional comments on the tables and figures.
There are several additional issues to be addressed.
-Figure 1 should be presented clearer to distinguish all lines, arrows and interceptions.
-The manuscript should be checked for the remaining technical text from the manuscript template.
-Author contributions should be presented.

-Abbreviation list is missing.

Author Response

Comment 1:  The presented paper is devoted to the comparative analysis of diagnostic significance of two biopsy methods: fine- 20 needle aspiration cytology (FNA) and core-needle biopsy (CNB). The authors concluded that CNB demonstrated the highest update rate to cancer.

Response: Thank you for your thoughtful comment. While we agree that a comparative analysis between FNA and CNB is an important component of the manuscript, we would like to clarify that this represents only part of the study’s scope. A substantial portion of the paper is further devoted to the evaluation of B3 lesions, including a detailed comparison of outcomes after vacuum-assisted excision (VAE) and any subsequent surgery. Our intention was to provide a comprehensive assessment that goes beyond the initial biopsy methods, focusing also on the diagnostic and therapeutic implications of managing B3 lesions.

Comment 2: The paper seems to be relevant to the scope of the journal. However, the results present only data analysis of moderate scale without any experimental work. Therefore, this manuscript could better fit to short communication format.

Response: We appreciate your positive assessment regarding the relevance of our work to the journal’s scope. Our intention was to provide a comprehensive evaluation based on the available dataset, which was derived from a prospective, single-blinded, randomized clinical trial comprised of 208 patients. We believe that this dataset offers meaningful insights to the field.

Comment 3: The subject area lacks novelty as comparison between fine- 20 needle aspiration cytology (FNA) and core-needle biopsy (CNB) were already published [PMID: 25174291]

Response: Thank you for your observation and for referring to the previously published comparison between FNA and CNB. We acknowledge that the diagnostic performance of these two biopsy methods has been explored in earlier studies. However, we would like to clarify that our manuscript extends beyond this comparison. In contrast to the cited article, our work places substantial emphasis on the management of B3 lesions, including outcomes following vacuum-assisted excision (VAE) —areas that were not addressed in the referenced study, which did not incorporate VAE and reported no B3 lesions (and only three C3 lesions).

Therefore, while part of our analysis involves FNA and CNB, the novelty of our study lies primarily in the comprehensive assessment of B3 lesions and their diagnostic and therapeutic pathways, which differentiates our work from the previously published data.

Comment 4: Authors should reformulate the conclusions to reach the necessary level of scientific novelty.

Response: We agree that clearly highlighting the scientific novelty of our work is important. While part of the manuscript addresses the comparison between FNA and CNB, the primary contribution of our study lies in the detailed analysis of B3 lesions, including the outcomes after VAE and subsequent surgery, areas that are detailed analysed in the discussion part of the manuscript.  We have now revised the conclusions to more explicitly emphasize these novel aspects and better reflect the added value of our findings within the field (last paragraph of the masnuscript, page 8-9, lines 264-272).

Comment 5: The manuscript lacks practical conclusions in, for example, routing of the patients with breast lesions in clinic.

Response:  We believe that this comment has been covered by the previous response.

Comment 8:  Figure 1 should be presented clearer to distinguish all lines, arrows and interceptions. The manuscript should be checked for the remaining technical text from the manuscript template. Author contributions should be presented. Abbreviation list is missing.

Response: Thank you for your remarks. The author contributions have been provided at the end of the manuscript and the Abbreviation list has been included as a supplementary file.

Reviewer 3 Report

Comments and Suggestions for Authors

This cohort study only demonstrates discrepancies among different diagnostic techniques—FNA, CNB, and VAE—in breast lesions based on categories B and C. Several points of criticism exist regarding this paper, even if multi-stage VAE could potentially serve as a surgical alternative to ADH in CNB. My comments are as follows.

  1. Generally, diagnostic techniques such as FNA, CNB, and VAE are determined based on the morphological characteristics and imaging findings of lesions identified by mammography (MG) and ultrasound (US), and these are used for cancer diagnosis. As is well known, each technique has several advantages and disadvantages. If lesion morphology is confirmed by MG and US, cancer diagnosis via FNA can be readily performed as an adjunct; however, VAE and CNB may sometimes be required to confirm tumor subtype. If FNA results are inconclusive (benign or malignant undetermined) and inconsistent with imaging findings, histopathological diagnosis via VAE or CNB is necessary depending on the required tissue sample volume. If the histopathological diagnosis still does not confirm cancer, surgical resection management proceeds with further diagnosis. Comparing imaging findings with histopathological findings is crucial for confirming a cancer diagnosis. As the authors pointed out, the lack of comparison between imaging findings and diagnostic techniques is a major criticism in this study.
  2. For lesions with risk factors where cancerous lesions coexist with ADH, FEA, and radial scars, management options such as observation, excision, or multiple VAE procedures during tumor board meetings may be considered. Since histopathological findings are key to determining the treatment plan, it is unlikely that multiple VAE procedures would be an inappropriate choice compared to other management approaches. How is the decision made during tumor board meetings to opt for multiple VAE procedures over other alternatives?
  3. Tumor heterogeneity is a common feature of breast cancer. The volume of tissue samples is a critical factor in confirming the histopathological diagnosis of breast tissue, particularly when determining whether it is cancerous. The selection of diagnostic techniques should be based on imaging features.
  4. When comparing final diagnoses made by VAE with postoperative diagnoses, why are pathological diagnoses in surgical specimens downgraded despite VAE having diagnosed cancer?
  5. Overall, the rate of disagreement in comparing diagnostic techniques is not significant; rather, it is necessary to analyze the causes of disagreement and the factors leading to it based on image features, histopathological findings, and tumor heterogeneity.

Author Response

Comment 1: As the authors pointed out, the lack of comparison between imaging findings and diagnostic techniques is a major criticism in this study.

Response: We agree with the reviewer’s observation. To mitigate this limitation as much as possible, we restricted our study population to patients whose imaging characteristics corresponded to BI-RADS categories 3 and 4a. This approach ensured a more homogeneous cohort with comparable imaging findings. We have now clarified this in the Materials and Methods section (lines 97–99).

Comment 2: For lesions with risk factors where cancerous lesions coexist with ADH, FEA, and radial scars, management options such as observation, excision, or multiple VAE procedures during tumor board meetings may be considered. Since histopathological findings are key to determining the treatment plan, it is unlikely that multiple VAE procedures would be an inappropriate choice compared to other management approaches. How is the decision made during tumor board meetings to opt for multiple VAE procedures over other alternatives?

Response : We would like to clarify that our study refers to multiple VAE rounds rather than multiple VAE procedures. In our technique, the target lesion was excised using VAE and placed entirely in the first container. Additional sampling from the surrounding area was then performed, and two separate containers were submitted to histopathology. Our aim was for the last container to contain no residual pathological tissue, thereby confirming the radicality of the excised area. This process is described in detail in the ‘Materials and Methods’ section (lines 105-110).

Comment 3: Tumor heterogeneity is a common feature of breast cancer. The volume of tissue samples is a critical factor in confirming the histopathological diagnosis of breast tissue, particularly when determining whether it is cancerous. The selection of diagnostic techniques should be based on imaging features.

Response: Thank you for pointing this out. The patients that are included in the study were having ultrasound-visible lesions of <30 mm in size, corresponding to imaging findings of BI-RADS 3 to 4a. This has been added to the revised version of the manuscript under the "Materials and Methods" "Trial populaion", page 3, line 97. as well as in the abstract.

Comment 4:  When comparing final diagnoses made by VAE with postoperative diagnoses, why are pathological diagnoses in surgical specimens downgraded despite VAE having diagnosed cancer?

Response : Thank you for your comment. We would like to clarify that VAE is intended to excise a lesion and therefore serves as part of the therapeutic algorithm, not only the diagnostic one. When VAE successfully removed a lesion and the subsequent surgery did not reveal any remaining pathology, the case was classified as a downgrade, thereby confirming the effectiveness and completeness of the VAE.

Comment 5: Overall, the rate of disagreement in comparing diagnostic techniques is not significant; rather, it is necessary to analyze the causes of disagreement and the factors leading to it based on image features, histopathological findings, and tumor heterogeneity.

Reponse: We agree that understanding the underlying causes of disagreement is essential. However, within the specific subset of lesions classified as BI-RADS 3 and 4a, our findings indicate that FNA demonstrates a higher discrepancy rate when compared with final histopathology.

Round 2

Reviewer 2 Report

Comments and Suggestions for Authors
  1. Tables 4 and 5 should be formatted according the Journal rules.
  2. Technical information on patents should be eliminated

Reviewer 3 Report

Comments and Suggestions for Authors

The authors appropriately addressed the reviewers' comments and revised the manuscript.